# Heart Failure Post-SARS-CoV-2 Infection in Children with Duchenne Muscular Dystrophy: The Additive Value of Cardiovascular Magnetic Resonance

**DOI:** 10.3390/children10050894

**Published:** 2023-05-17

**Authors:** George Markousis-Mavrogenis, Antonios Belegrinos, Aikaterini Giannakopoulou, Evangelos Karanasios, Vasiliki Vartela, Elephtheria Patsilinakou, Paul Samantis, Roser-Marie Pons, Antigoni Papavasiliou, Sophie I. Mavrogeni

**Affiliations:** 1Onassis Cardiac Surgery Center, 17674 Athens, Greece; 2Medical School, National and Kapodistrian University of Athens, 11527 Athens, Greece; 3Aghia Sophia Children’s Hospital, 11527 Athens, Greece; 4Olympic Diagnostic Research Center, 18543 Piraeus, Greece; 5Iaso Children’s Hospital, 15123 Athens, Greece; 6University Research Institute of Maternal and Child Health and Precision Medicine, National and Kapodistrian University of Athens, 11527 Athens, Greece; 7Center for Adolescent Medicine and UNESCO Chair in Adolescent Health Care, National and Kapodistrian University of Athens, 11527 Athens, Greece

**Keywords:** cardiovascular magnetic resonance, COVID-19, Duchenne Muscular Dystrophy, myocarditis

## Abstract

In this case series, we describe the diagnosis of post-COVID-19 myocarditis in asymptomatic patients with Duchenne Muscular Dystrophy (DMD) and a mild COVID-19 disease course. These patients were referred for CMR due to electrocardiographic and echocardiographic alterations, which did not exist before COVID-19 infection. CMR identified the presence of severe myocardial inflammation in all patients based on abnormally elevated myocardial T2 ratio, late gadolinium enhancement, native T1 mapping, T2 mapping, and extracellular volume fraction. This was paired with concurrent impairment of left ventricular function. Appropriate treatment was initiated in all cases. Two of the four patients developed episodes of ventricular tachycardia during the following 6 months, and a defibrillator was implanted. Despite the mild clinical presentation, this case series demonstrates the diagnostic strength of CMR in the diagnosis and evaluation of post-COVID-19 myocarditis and serves to increase awareness of this potential complication amongst treating physicians.

## 1. Introduction

Neuromuscular disorders (NMDs) are a heterogeneous group of conditions, characterized by progressive muscle degeneration and weakness. Cardiac and respiratory dysfunction are common findings that can severely affect the clinical course of the disease [1].

To our knowledge, whether patients with NMDs are at an increased risk of worse outcomes of coronavirus disease 2019 (COVID-19) has not yet been investigated. Factors that have been linked to worse COVID-19 outcomes are Forced Vital Capacity (FVC) < 60%, the requirement for invasive or non-invasive ventilation, insufficient airway clearance due to oropharyngeal weakness, and systemic comorbidities, such as heart disease, diabetes, obesity, and iatrogenic immunosuppression [1,2]. Notably, fever often accompanying COVID-19 can worsen muscular dysfunction or trigger rhabdomyolysis in some types of NMDs, such as metabolic myopathies and myasthenia gravis, thus resulting in worse outcomes [3,4]. Furthermore, in patients with poorly adjusted corticosteroid dosage, fever might also precipitate an adrenal crisis [3,4].

In this case series, we present four pediatric patients with Duchenne muscular dystrophy (DMD), who developed rapid left ventricular deterioration after COVID-19 infection. By doing so, we aim to increase awareness of this possible complication in this patient population.

## 2. Case Presentation

Four non-ambulatory DMD patients without cardiorespiratory symptoms, aged 6–10 years, underwent cardiac evaluation <1 month after a diagnosis of coronavirus disease 2019 (COVID-19) and resolution of symptoms to rule out the presence of potential sequelae of COVID-19 that might complicate the underlying DMD diagnosis. The main symptoms at COVID-19 diagnosis were sore throat, headache, fever lasting less than three days, cough, myalgia, and anosmia, and all patients were under treatment with corticosteroids for DMD. None required hospitalization for COVID-19, and none experienced serious clinical complications as a consequence thereof. Additionally, none were vaccinated because they were younger than 12 years old.

Clinical evaluation revealed an irregular peripheral pulse in all cases, which was followed by 12-lead electrocardiography (ECG). One patient presented short episodes of supraventricular and ventricular runs (3–4 beats). A 24 h Holter recording revealed runs of ventricular tachycardia (VT) in two of the patients. Echocardiography was additionally performed and revealed decrements in left ventricular ejection fraction (LVEF) in 2/4 cases, compared with the immediately preceding screening echocardiogram routinely performed at 1-year intervals. Thus, all patients were referred for additional evaluation using cardiovascular magnetic resonance imaging (CMR) to determine the presence of fibrosis or investigate the new reduction in systolic function.

CMR was performed within the first month (30 ± 5 days) after the resolution of COVID-19 symptoms using a 3.0 T scanner (Magnetom Skyra, Siemens Healthcare, Erlangen, Germany) and a 32-channel phased-array receiver coil. The study protocol has been published previously [5]. Briefly, we evaluated biventricular volumes, ejection fractions, and myocardial oedema using short-tau inversion recovery (STIR) T2-weighted imaging and T2 mapping, as well as myocardial inflammation/fibrosis using early and late gadolinium-enhanced images (EGE/LGE) after contrast medium injection, native (pre-contrast) T1 mapping, post-contrast T1 mapping, and extracellular volume fraction (ECV). The presence of myocardial inflammation was documented using the updated Lake Louise criteria [6]. Locally used cut-off points for abnormal values based on data from healthy volunteers were as follows: native T1 mapping > 1200 ms, ECV > 28%, T2-ratio > 2, EGE > 4, LGE > 0% of LV mass, and T2 mapping > 50 ms.

The CMR findings of all cases are presented in Table 1. We observed left ventricular dilatation in all patients, with moderate to severe reduction in LVEF, while three out of four patients only showed mild decrements in RVEF, and one showed a severe reduction. Notably, all patients showed evidence of severe myocardial edema, hyperemia, and fibrosis, exemplified by increased T2 ratio, EGE, and LGE values, respectively. LGE was localized in the inferolateral LV in three out of four patients and in the interventricular septum, anterior, lateral, and inferior LV walls in Case 1. Illustrative examples are presented in Figure 1 and Figure 2. Parametric CMR indices further confirmed the presence of active myocardial inflammation with high probability based on the updated Lake Louise criteria, as all patients showed pathologic values for native T1 mapping, T2 mapping, and ECV. A diagnosis of post-COVID-19 myocarditis with severe LV dysfunction was made in all cases.

Treatment with angiotensin-converting enzyme inhibitors and β-adrenoreceptor antagonists was initiated in all patients. Due to the absence of signs and symptoms of heart failure, all patients were treated on an outpatient basis. After 6 months, cases one and four developed multiple episodes of VT (Figure 3 and Figure 4). None of the patients had polymorphic VT or indications of different origins of ectopy. Due to the multiple daily occurrences of non-sustained VT alongside the reduced LVEF, these patients additionally received amiodarone (2.5 mg per kg of body weight every 24 h), and the decision was made to implant a cardioverter defibrillator. The two younger patients were ambulatory and were followed up for 6 months with exercise limitation. At the time of writing, approximately 2 years after the CMR examination, all patients remained alive and had a similar quality of life and exercise capacity compared with the time before which they experienced COVID-19.

## 3. Discussion

In this case series, we present four pediatric patients with DMD and previously normal LV function, who developed post-COVID-19 myocarditis with severe LV dysfunction. To our knowledge, all previous studies that investigated the clinical presentation of COVID-19 in patients with DMD concluded that its clinical manifestations remain mild, and all patients made a complete recovery [7,8,9,10] However, it should be noted that none of the aforementioned studies made use of any cardiac imaging. The novelty of our study is that all patients were thoroughly evaluated using a standard cardiology examination, ECG, 24 h Holter, echocardiography, and CMR. Despite the lack of cardiac symptoms in any of the cases we examined, preliminary cardiac examination noted some abnormalities, which were later substantiated by CMR, which demonstrated extensive cardiac inflammation and fibrosis, seemingly out of proportion with patient symptoms. Thus, it could be argued that standard clinical examinations and history taking may be insufficient for uncovering the complete extent of organ damage that could be caused by COVID-19. An additional possibility is the presence of multisystem inflammatory syndrome in children (MIS-C). MIS-C is a systemic inflammatory disease that affects children and adolescents infected by SARS-CoV-2. Although there are various definitions of this syndrome, all of them include either the presence of fever for at least three days or more or the involvement of multiple organ systems, which were not detected in any of the patients in our case series [11].

The findings presented here are important because of the previously reported observation that dystrophin deficiency increases myocardial susceptibility to viral infections. Dystrophin-deficient mice that were infected with coxsackievirus B3 displayed greater viral replication and more severe cardiomyopathy compared with wild-type mice with normal dystrophin [12]. In humans, mutations in cytoskeletal proteins have also been associated with an increased severity of myocarditis [13]. It is possible that myocardial viral infections could result in the progression of DMD to hypertrophic or dilated cardiomyopathy through an interaction between viral proteases and cytoskeletal proteins [13]. Similar phenomena secondary to myocardial SARS-CoV-2 infection might explain the ventricular dilation, cardiac inflammation, and impaired contractile function observed in our patients. However, the precise underlying mechanisms remain to be elucidated.

Currently, there is, to our knowledge, no published literature regarding the role of CMR in patients with DMD in the convalescent phase after COVID-19. However, CMR revealed cardiac involvement in 78% and ongoing myocardial inflammation in 60% of patients who recently recovered from COVID-19, independent of various relevant covariates [10]. Our preliminary findings support the more extensive use of CMR in these patients, particularly in the presence of ECG and/or echocardiographic abnormalities.

Our findings have important clinical implications for the management of patients with DMD who have recently recovered from COVID-19. Our study stresses the fact that a seemingly mild disease course of COVID-19 does not rule out the possibility of severe underlying myocardial inflammation. In particular, patients with DMD who are not ambulatory and require the use of a wheelchair might not perceive typical symptoms such as shortness of breath, chest pain, or palpitations due to their limited mobility. As such, a high index of suspicion is required on the part of the treating physicians. Furthermore, the early identification of myocardial inflammation can motivate the initiation of cardioprotective treatment and more vigilant clinical monitoring, as was done in all cases we examined.

### Limitations of the Study

Our study has the following limitations:Given the relatively small number of patients, the true incidence of cardiac involvement post-COVID-19 in patients with DMD cannot be accurately estimated. Additionally, comparisons between patients with and without DMD are necessary in order to determine whether this patient subgroup is at greater risk of COVID-19-related cardiac injury.No previous CMR examination was available, which limited our ability to infer changes over time, secondary to SARS-CoV-2 infection.Lastly, there was no long-term follow-up available, and thus, no comments could be made regarding the potential effects of CMR findings on prognosis.

## 4. Conclusions

In our case series of patients with DMD who recently recovered from COVID-19, we identified the presence of myocardial inflammation/fibrosis with deleterious effects on myocardial function despite the lack of cardiac symptoms and a severe COVID-19 disease course. Our results support the more widespread use of CMR in this patient population and serve to raise awareness amongst treating physicians regarding this severe potential complication.

## Figures and Tables

**Figure 1 children-10-00894-f001:**
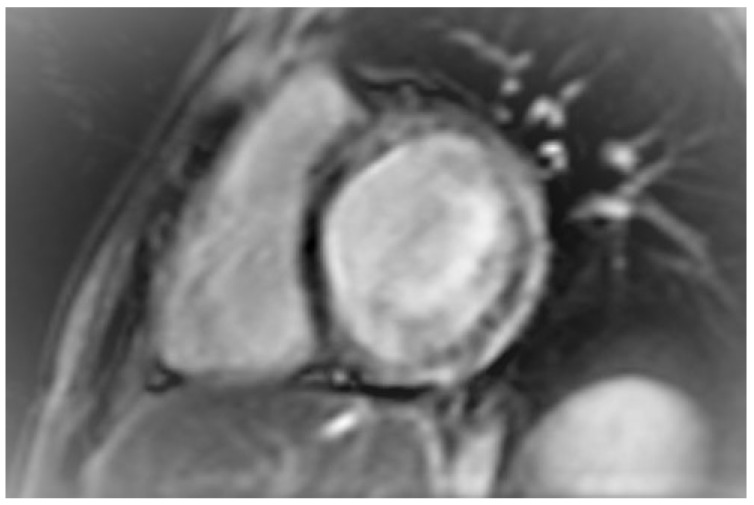
Short axis inversion recovery image showing late gadolinium enhancement in the interventricular septum, and the anterior, lateral, and inferior wall of the left ventricle (Case 1).

**Figure 2 children-10-00894-f002:**
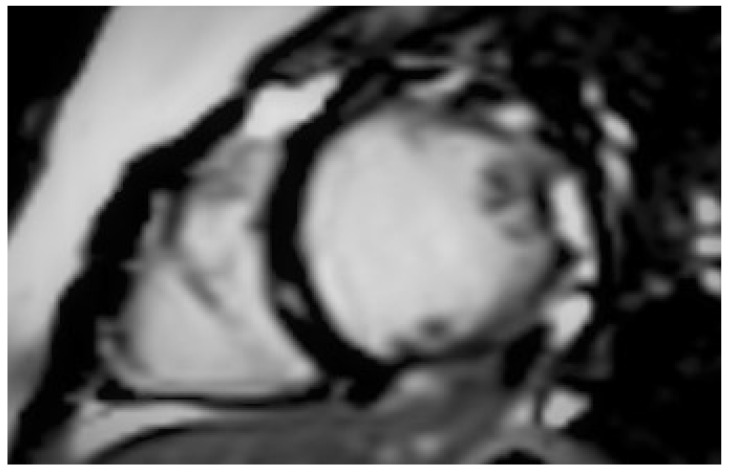
Short axis inversion recovery image showing late gadolinium enhancement in the inferolateral wall of the left ventricle (Case 3).

**Figure 3 children-10-00894-f003:**
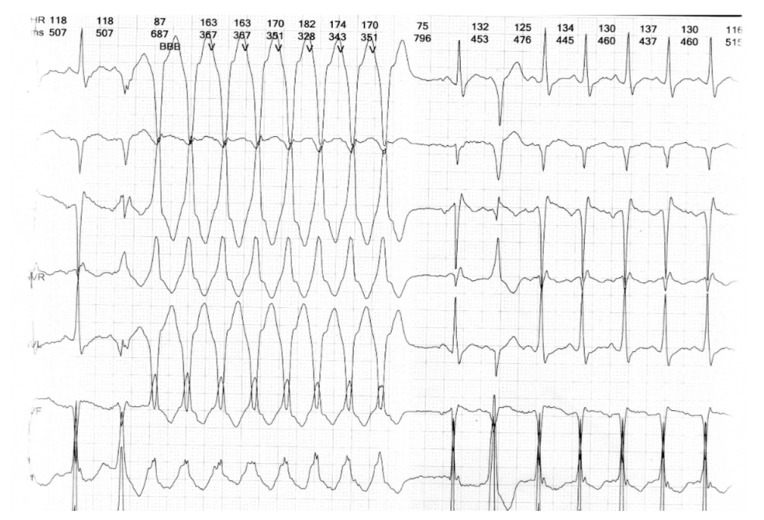
24 h Holter recording showing non-sustained ventricular tachycardia (Case 1).

**Figure 4 children-10-00894-f004:**
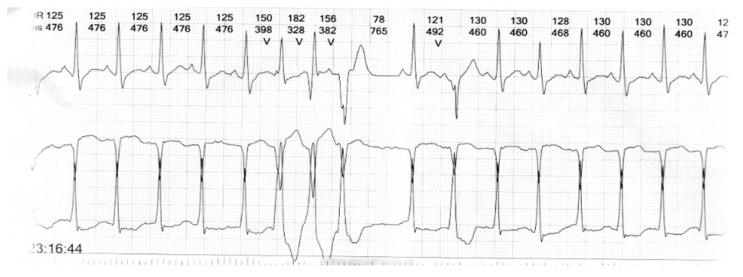
24 h Holter recording showing non-sustained ventricular tachycardia (Case 4).

**Table 1 children-10-00894-t001:** CMR findings in the four cases of patients with Duchenne muscular dystrophy. Cut-off points for pathologic values were T1 mapping > 1200 ms, ECV > 28%, T2-ratio > 2, EGE > 4, LGE > 0% of LV mass, and T2 mapping > 50 ms.

Patient	LVEDV (mL)	LVESV (mL)	LVEF (%)	RVEDV (mL)	RVESV (mL)	RVEF (%)	T2 Ratio	EGE	LGE (% of LV Mass)	LGE Localisation	Native T1 Mapping (ms)	T2 Mapping (ms)	ECV (%)
Case 1	166	126	24	73	41	43	2.4	5	28	IVS, anterior, lateral, and inferior LV	1350	55.5	32
Case 2	111	76	38	61	30	46	2.7	15.5	25	Inferolateral LV	1557	53.5	44
Case 3	100	53	46	73	48	34	2.8	6	18	Inferolateral LV	1380	52	30.5
Case 4	135	102	34	43	25	41	2.6	8	18	Inferolateral LV	1370	59.5	32.7

CMR, cardiovascular magnetic resonance; LV/RV, left/right ventricular; EDV/ESV, end-diastolic/end-systolic volume; EF, ejection fraction; EGE/LGE, early/late gadolinium enhancement; ECV, extracellular volume fraction; IVS, interventricular septum.

## Data Availability

Complete data are not provided in this case, in order to protect patient safety. Data can be made available upon reasonable request to the corresponding author.

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
