# Peer review of "Heart Failure Post-SARS-CoV-2 Infection in Children with Duchenne Muscular Dystrophy: The Additive Value of Cardiovascular Magnetic Resonance"

_children, 2023, doi:10.3390/children10050894_

Round 1
Reviewer 1 Report
The author describes 4 cases of DMD with post-Covid myocarditis which is a very interesting topic.
The title is "heart failure post-covid..." whereas the article focuses on MRI findings in patients with no heart failure symptoms but gives a very detailed description of the MRI protocol in the methodology section. I suggest to change this in the title.
The abstract should be rewritten with some more details regarding the MRI results (LVEF, LGE,..).
Methods: Generally the methods are well described. Below some comments:
1. Symptoms during Covid-19 infection should be mentionned at the begining. Otherwise, it is not clear from the text, wether these symptoms were still present at one month post-Covid.
2. What were the reasons to perform an echocardiography one month after Covid-19 if the patient was asymptomatic? Could you please give an explanation.
3. How long after the Covid-19 infection was the MRI performed?
Results:
1. To enhance the reading of the article and to allow a better comparison between the 4 cases it is suggested to present the MRI results in a table.
2. Could you please provide some more details about the VT? Was it polymorphic VT suggesting different origins of the ectopy? Did the patient receive any anti-arrythmia medication besides the defibrillator?
3. Two patients had severe LV dysfunction. Did they receive iv milrinone or other therapy in addition to ACE inhibitors and beta-blockers?
Discussion:
Another explanation for heart failure post-Covid 19 in children is PIMS-TS. This should be discussed.
Limitations of the study should be explained more in details.
Author Response
We appreciate the reviewers suggestions and made changes accordingly. Specificaly, the changes made are;
1) changed title to ;
Reviewer 2 Report
An interesting and educational topic that has clinical merit; however, there are editing issues that the authors should consider and address. The following are suggestions/comments regarding those issues. Line 21, "... ventricular tachycardia during the next 6 months ...". Line 34, "... there has not been an established link between ....". Line 35, " .... higher risk of Covid-19. Factors that have ...". Line 39, "medications (1-3). In some subgroups ...". Line 40, "... common symptom of Covid-190, can trigger muscle ...". Line 43, "...contributing to a poor prognosis in ...". Line 53, "... volume fraction (ECV), it is possible to ...". Lines 56 & 57, "...evaluation and developed fibrosis more ...". Line 61, " ... were referred for CMR imaging, due to the ...". Line 65, " ... while the remaining patient presented with short episodes of ...". Lines 66 & 67, "...in all of them and no patient was taking any cardiac medication(s)." Line 68, "... neither at the time of the Covid-19 infection nor at the ...". Line 69, "...main symptoms at presentation in the patients ...". Line 70, "... of the infected patients required hospitalization ....". Line 79, "... study was considered positive, if 2 out 3 ...". Line 128, "...T2-mapping values of the myocardium." Line 157, "... enhancement (LGE) =28% of the LV mass, involving the ....". Line 158, "...lateral and inferior walls of the LV." Line 162, "...subepicardial LGE = 25% of the LV mass ...". Line 163, "...and inferior walls of the LV." Line 166, "...subepicardial LGE = 18% of the LV mass ...". Line 167, ".... inferior walls of the LV." Line 170, ".... subepicardial LGE =18% of the LV mass ....". Line 171, "... and inferior walls of the LV." Line 174, "After 6 months, case 1 and 4 developed ...". Lines 187 & 188, "These affected patients suffered from greater ...". Line 188 & 189, "...pulmonary function and lung disease, stemming from ...". Lines 192 & 193, "...that could possibly contribute to the severeness of the ...". Line 198, "obese, treated with corticosteroid medication ...". Line 199, "...29 children with NMDs infected by Covid-19." Line 203, "...DMD patients was mild and unrelated to serious ...". Line 206, "...our patients also had a mild ...". Line207 & 208, "However, echocardiography identified ...". Line 208, "...Covid-19 infection, followed with a CMR, which identified the ...". Lines 209 & 210, "... extensive fibrosis that could not be justified by the ...". Line 212, "factors except for the use of ....". Lines 216 & 217, "...in DMD patients. Line 219, "evaluation of DMD patients post Covid-19 ...". Line 220, "...CMR in DMD patients post Covid-19 ...". Lines 223 & 2224, "...from the original diagnosis, emphasizing the need for ...". Line 226, "Covid-19 infection ...". Lines 226 & 227, "... myocarditis in DMD patients, our first findings ...". Line 229, "... potentially useful as a selection criteria for ...". Line 235, "...treatment and potentially eliminate future serious complications." Line 244, "... with deleterious effects on myocardial ...". Line 248, "a guide for the initiation of cardiac treatment."
I would also recommend capitalizing COVID-19 throughout the manuscript.
Author Response
We thank the reviewer for their contribution. We tried to make improvements throughout the article
Reviewer 3 Report
Methods
In the first line of the methods section, the manuscript states ' 4 DMD patients without cardiorespiratory symptoms...'
5 lines later (line 64), the manuscript states 'three out of four patients were free of cardiac symptoms, while the rest one presented short runs of SVT and ventricular runs'
It is not clear whether the manuscript means that following Covid infection, one patient developed cardiac arrhythmias that were not present before. This should be better defined. Also the grammar is incorrect in line 64/65 and makes the sentence difficult to understand.
The rest of the methods section is explained very well.
Results
Obviously small patient number but considering the rarity of the disease and the importance of the findings, this is an acceptable limitation.
In each patient's section, it would be interesting to know their baseline echo function and the changes seen post Covid infection which prompted referral for MRI, plus any ECG changes noted. This would be useful in order to know what changes should precipitate referral for CMR.
Were the runs of ventricular tachycardia unifocal or multifocal and were they detected on holter recordings? Were they associated with symptoms such as palpitations or syncope and had the patients had previous normal holter recordings prior to their Covid 19 infection?
I think these are important points that require clarification.
The fact these patients had no previous MRI to compare to is a limitation and should be mentioned.
Author Response
Dear reviewer,
Thank you for your input. We tried to respond to your comments in a satisfactory manner. Specifically;
Methods
In the first line of the methods section, the manuscript states ' 4 DMD patients without cardiorespiratory symptoms...'
5 lines later (line 64), the manuscript states 'three out of four patients were free of cardiac symptoms, while the rest one presented short runs of SVT and ventricular runs'
Major changes have been made to the text, taking that into account
It is not clear whether the manuscript means that following Covid infection, one patient developed cardiac arrhythmias that were not present before. This should be better defined.
We tried to clarify it along with the changes
Also the grammar is incorrect in line 64/65 and makes the sentence difficult to understand.
We tried to clarify it along with the changes
The rest of the methods section is explained very well.
Results
Obviously small patient number but considering the rarity of the disease and the importance of the findings, this is an acceptable limitation.
In each patient's section, it would be interesting to know their baseline echo function and the changes seen post Covid infection which prompted referral for MRI, plus any ECG changes noted. This would be useful in order to know what changes should precipitate referral for CMR.
We tried to make clearer what changes and observations lead each patient to MRI investigation
Were the runs of ventricular tachycardia unifocal or multifocal and were they detected on holter recordings? Were they associated with symptoms such as palpitations or syncope and had the patients had previous normal holter recordings prior to their Covid 19 infection?
The tachycardia was unifocal. We added more information regarding the tachycardia in the text, explaining its charachteristics
I think these are important points that require clarification.
The fact these patients had no previous MRI to compare to is a limitation and should be mentioned.
It was added in the limitations section Kind regards, the authorsReviewer 4 Report
Overall this is an interesting case series describing LV systolic dysfunction after COVID infection. I have some suggestions regarding edits:
1. The introduction, COVID 19 is not clearly defined and is used interchangeably with covid. Please define clearly and only utilize one style of wording. The definition comes in the discussion, when it should have been in the introduction.
2. The authors mention the use of T1 mapping and ECV in the methods, when discussing the CMR protocol for myocarditis but it does not appear that these sequences were used as part of the criteria for defining myocarditis (the methods mention EGE, LGE and T2 weighted images). Can you clarify then what role the T1 mapping and ECV played?
3. The presentation of the results is difficult to read - I would suggest just moving all of the values to a table and only mention the highlights in the results instead of typing out all findings.
4. Patients 1 and 4 are briefly mentioned to have developed ventricular arrythmias six months post covid - it seems unusual that this would be due to covid alone. Did these patients have other factors or persistently depressed ventricular function? Could you include follow up findings all four patients six months post covid, perhaps in a table?
5. The second paragraph of the discussion feels more appropriate for the introduction than discussion.
6. The authors acknowledge that larger scale studies have shown minimal to no covid complications in this patient population - might there be other factors that caused these four patients to develop worsening heart failure? This is not adequately explored in the discussion. It might be helpful to include ECG data etc from the time of covid diagnosis and include demographic data in a table.
7. There are several typos and grammatical errors that need to be fixed.
Author Response
Dear reviewer,
We thank you for your input. We tried to make satisfactory changes. Specifically
- The introduction, COVID 19 is not clearly defined and is used interchangeably with covid. Please define clearly and only utilize one style of wording. The definition comes in the discussion, when it should have been in the introduction.
We chose to use Covid 19 thorough the text and standardize its use in it.
- The authors mention the use of T1 mapping and ECV in the methods, when discussing the CMR protocol for myocarditis but it does not appear that these sequences were used as part of the criteria for defining myocarditis (the methods mention EGE, LGE and T2 weighted images). Can you clarify then what role the T1 mapping and ECV played?
The part describing the use of MRI was rewritten in a clearer maner
- The presentation of the results is difficult to read - I would suggest just moving all of the values to a table and only mention the highlights in the results instead of typing out all findings.
The results were presented in a table format in order to increase readability
- Patients 1 and 4 are briefly mentioned to have developed ventricular arrythmias six months post covid - it seems unusual that this would be due to covid alone. Did these patients have other factors or persistently depressed ventricular function? Could you include follow up findings all four patients six months post covid, perhaps in a table?
Follow up is not currently possible and addressed in limitations. We believe that apart from the covid 19 infection and their baseline disease (DMD) no other factor was to blame for the arrythmias. The possibility of MIS-C was addressed.
- The second paragraph of the discussion feels more appropriate for the introduction than discussion.
We believe that after the changes made in the text, this was adressed
- The authors acknowledge that larger scale studies have shown minimal to no covid complications in this patient population - might there be other factors that caused these four patients to develop worsening heart failure? This is not adequately explored in the discussion. It might be helpful to include ECG data etc from the time of covid diagnosis and include demographic data in a table.
We tried to discuss the possibility of MIS-C in the discussion. The sudden changes to what would be otherwise stable patients that lead some of the children to investigation along with the time that passed between the infection and the findings lead us to believe that the involvement of Sars-CoV 2 was the causal factor. We included the ECG in the paper
7. There are several typos and grammatical errors that need to be fixed.
Multiple changes were made
Round 2
Reviewer 1 Report
Please check the layout of the tables.
Author Response
Dear reviewer,
We thank you for your input. We will make the table as readable as possible
Kind regards
Reviewer 4 Report
The authors have undertaken significant improvements in this manuscript, which now improved. However, the formatting of the tables appears to require some work before it is ready for publication. I would recommend sending a clean copy with the tables formatted appropriately.
Author Response

(The authors gave the same response as above.)
